# The Stability and Anti-Angiogenic Properties of Titanium Dioxide Nanoparticles (TiO_2_NPs) Using Caco-2 Cells

**DOI:** 10.3390/biom12101334

**Published:** 2022-09-21

**Authors:** Oladipupo Moyinoluwa David, Kim Leigh Lategan, Maria Fidalgo de Cortalezzi, Edmund John Pool

**Affiliations:** 1Department of Medical Bioscience, University of the Western Cape, Cape Town 7535, South Africa; 2Department of Civil and Environmental Engineering, University of Missouri, Columbia, MO 65211, USA

**Keywords:** titanium dioxide, nanoparticles, cytotoxicity, inflammation, cell stress, angiogenesis

## Abstract

Titanium dioxide nanoparticles (TiO_2_NPs) are found in a wide range of products such as sunscreen, paints, toothpaste and cosmetics due to their white pigment and high refractive index. These wide-ranging applications could result in direct or indirect exposure of these NPs to humans and the environment. Accordingly, conflicting levels of toxicity has been associated with these NPs. Therefore, the risk associated with these reports and for TiO_2_NPs produced using varying methodologies should be measured. This study aimed to investigate the effects of various media on TiO_2_NP properties (hydrodynamic size and zeta potential) and the effects of TiO_2_NP exposure on human colorectal adenocarcinoma (Caco-2) epithelial cell viability, inflammatory and cell stress biomarkers and angiogenesis proteome profiles. The NPs increased in size over time in the various media, while zeta potentials were stable. TiO_2_NPs also induced cell stress biomarkers, which could be attributed to the NPs not being cytotoxic. Consequently, TiO_2_NP exposure had no effects on the level of inflammatory biomarkers produced by Caco-2. TiO_2_NPs expressed some anti-angiogenic properties when exposed to the no-observed-adverse-effect level and requires further in-depth investigation.

## 1. Introduction

Titanium dioxide nanoparticles (TiO_2_NPs) exist in various forms, of which anatase, brookite and rutile are the most abundant. The rutile form has been studied extensively due to its electrical, optical and thermal properties [1]. Titanium dioxide nanoparticles are white pigments, and due to their brightness and high refractive index, they have a variety of applications [2]. These are sought-after characteristics, resulting in TiO_2_NPs being one of the earliest industrially and most abundantly produced nanoparticles globally [3]. Applications of TiO_2_NPs include paints, food products, cosmetics, toothpastes, plastics, industrial photocatalytic processes and, very commonly, sunscreens, as it helps protect the skin from UV light [4,5,6].

Recent reports indicated that TiO_2_NPs might cause adverse environmental effects. Due to the high presence of these NPs in consumer products, humans can also potentially be adversely affected [7,8,9]. The in vivo effects of TiO_2_NP exposure have indicated acute toxicity of a number of organs such as lung, kidney and liver, and as well as immune toxicity [5,10,11]. However, there are conflicting reports showing that TiO_2_NPs are not significantly absorbed in the liver, kidneys or small intestine [10,11]. The main route of excretion of TiO_2_NPs is via the urinary tract [9,12]. Contradictory reports regarding in vitro toxicity of TiO_2_NPs also exist in which some report genotoxicity and cytotoxicity in some cell lines, while others do not find adverse effects [13,14,15,16,17,18,19,20,21]. The possible reasons for these inconsistent reports are the use of different cell types, animal models, doses and sizes of the nanoparticle used.

Reviews of the in vitro toxicology of TiO_2_NPs for mammalian cells indicated that there were few studies on the effects of these NPs on intestinal cells, the major intestinal barrier of the immune system and the body. It was also noted that the NPs could cross the intestinal epithelium layer via transcytosis without damaging the epithelial cell integrity but resulting in minor effects [3,8].

To the best of our knowledge, limited studies have assessed how these NPs behave when in physiological media containing serum constituents. Serum is thought to affect the aggregation and stability of the NP. Fatisson et al. (2012) noted that engineered NPs (ENPs) were only moderately affected by cell culture media but that in the presence of serum, the NPs were destabilized after 24 h [22]. Another study noted an increase in NP size when in cell culture media, and these factors can attribute to the conflicting reports of TiO_2_NP toxicity in vivo and in vitro [23].

The current study aimed to evaluate the effects of various physiological media on TiO_2_NP hydrodynamic size and zeta potential. In addition, the physiological effects of these characterized TiO_2_NPs on cell viability, cell stress and inflammatory biomarkers were investigated. Our rationales for the use of biomarker profiles were that these would provide rapid information on potential adverse and beneficial effects of TiO_2_NPs for a large number of conditions that could potentially be affected by the NPs and also give indications of pharmaceutical activity that can potentially be used as intervention therapies for various diseases.

## 2. Materials and Methods

### 2.1. Nanoparticle Characterization

The aeroxide P_25_ TiO_2_NPs were provided by the manufacturer (Evonik Degussa Corporation) CAS: 13463-67-7 and the manufacturers reported a spherical shape and an average primary particle size of 21 nm as a hydrophilic fumed TiO_2_ mixture of rutile and anatase forms. The TiO_2_NPs were distributed by transmission electron microscope (TEM) to determine the morphology and size of the TiO_2_NPs. Scanning electron microscope (SEM) with energy-dispersive x-ray was used to confirm elemental titanium shape and size. X-ray diffraction (XRD) was performed with a Philip expert pro MPD X-ray diffractometer using Cu-k radiation at voltage 40 kV and current 40 mA to determine the structure of the TiO_2_NPs. The TiO_2_NPs were subsequently characterized in various media over a 2-week period to determine whether their characteristics would alter. Thereafter, the TiO_2_NPs were placed in various media (pH 7): 150 mM sodium chloride (NaCl) (Sigma-Aldrich, St. Louis, MO, USA); 1× phosphate buffered saline (PBS) (Sigma-Aldrich); Incomplete Dulbecco’s Modified Eagle’s Medium (DMEM) (Sigma-Aldrich) containing 0.1% glutamax (Sigma-Aldrich), 0.1% antibiotic/antimycotic solution (Sigma-Aldrich) and 0.05% gentamicin (Sigma-Aldrich); and complete DMEM media, containing the same constituents as incomplete media but containing additional 10% heat inactivated foetal bovine serum (FBS) (Sigma-Aldrich) to yield a final concentration of 10 µg/mL TiO_2_NPs. The TiO_2_NPs in the various medias were then incubated at 37 °C for 0 h, 24 h, 7 and 14 days respectively. After the various incubation periods, the medium was centrifuged at 10,000 rpms (Eppendorf Centrifuge 5810 R), and the NP pellets were then washed and resuspended in dH_2_0. Thereafter, disposable folded capillary cells (Malvern Pananlytical) were used to determine the hydrodynamic size and zeta potential of the nanoparticles with a ZetaSizer Nano ZS (Malvern Instruments, Malvern, PA, USA).

### 2.2. Nanoparticle Preparation for Cell Culture

A 10 mg/mL stock of TiO_2_NPs was prepared in distilled water. The NPs were sonicated (QSonica, LLC. Misonixsonicators, XL-200 Series) on ice for short burst for approximately 5 min. Nanoparticles were freshly prepared prior to each experiment.

### 2.3. Cell Culture

The human colorectal adenocarcinoma (Caco-2) epithelial cell line was obtained from the American Type Culture Collection (ATCC HTB-37). Standard tissue culture conditions were used to maintain the cells in complete medium. The Caco-2 cells were cultured in Dulbecco’s modified Eagle’s medium (DMEM, Lonza, Cape Town, South Africa) supplemented with 10% heat inactivated fetal bovine serum (FBS, Hyclone, Little Chalfont, UK), gentamicin, glutamax and antibiotic/antimycotic (Sigma-Aldrich, St. Louis, MO, USA). The cells were incubated in a humidified atmosphere of 5% CO2 at 37 °C. Cells were sub-cultured approximately every 3–4 days using 0.05% trypsin ethylenediaminetetraacetic acid (EDTA) (Gibco). Caco-2 cells were seeded at a density of 4 × 10^3^ cells/mL in 24-well tissue culture-treated plates (Nunc) and were approximately 60% confluent before nanoparticle treatment. Cells were exposed for 48 h to 0–500 µg/mL TiO_2_NPs, respectively as inflammatory markers could not be detected after 24 h exposure. Thereafter, supernatants removed and cell viability assessed. Supernatants were centrifuged at 12.1 rcfs for 1 min (MiniStar Plus Super Mini Centrifuge) before evaluating innate inflammatory biomarkers and performing an angiogenesis proteome profile. Experiments were repeated and cells were harvested and protein concentration quantified in order to evaluate cell stress biomarkers potentially induced by the NPs.

### 2.4. Cell Viability Assay

Cell viability was monitored using the sodium 3′[1-[(phenylamino)-carbony]-3,4-tetrazolium]-bis(4-methoxy-6-nitro) benzene-sulfonic acid hydrate) (XTT) assay (Sigma-Aldrich). The assay monitors the conversion of the XTT tetrazolium salt to a soluble formazan salt in metabolically active cells [24]. Therefore, an increase in the conversion to formazan is directly proportional to cell viability. Supernatants were removed and cells washed with PBS to remove any excess NP that may interfere with the assay. A 1:50 ratio of XTT coupling reagent to XTT labelling reagent was prepared. This mixture was further diluted in complete medium to yield a final ration of 1:3. Plates were immediately read at 450 nm (FLUOstar Omega, BMG Labtech, Ortenberg, Germany) after the addition of XTT. Plates were then incubated at 37 °C for 1 h after which an additional reading was made. The change of absorbance over time was calculated and the percentage viability analysed.

### 2.5. NO Assay

The amount of nitrite produced by the Caco-2 cells exposed to the respective nanoparticle concentrations was assessed in the cell culture supernatant as an indication of NO production. This assay is based on the Griess reaction [25]. The amount of nitrite produced by the cells was measured against the nitrite standard range (Sigma-Aldrich) (0–100 µM). A 1:1 of culture supernatant or standard was mixed with the Griess reagent (1:1 of 1% sulphanilamide and 0.1% naphtylethlemidimine-dihydrochloride in 2.5% sulphuric acid (all obtained from Sigma-Aldrich). The plate was subsequently read at 540 nm (FLUOstar Omega, BMG Labtech) and the amount of nitrite produced by the cells quantified.

### 2.6. Innate Inflammatory Biomarkers IL-6 and IL-8

Double antibody sandwich enzyme linked immuno-sorbent assays (DAS-ELISA) were run according to the manufacturer’s instructions. Both the IL-6 (Invitrogen) and IL-8 (R & D Systems) assays were run using undiluted supernatants.

### 2.7. Protein Quantification

The cells were harvested after NP exposure using lysis buffer solution (1× PBS, 0.1% tween and 200 µL protease inhibitor). The cells were then scraped with a scrapper after which cells were then sonicated (QSonica, LLC. Misonixsonicators, XL-200 Series) on ice for short burst for 20 secs and centrifuged at 12.1 rcfs for 1 min. Protein concentration of the cell homogenate was quantified using Bradford reagent. The cell homogenates at 300 µg/mL protein were used to evaluate cell stress biomarkers potentially induced by the nanoparticle.

### 2.8. Cell Stress Biomarkers

The production of cell stress biomarkers induced by exposing the cells to the nanoparticles was assessed by performing SOD, Phospho-HSP27 and HSP70 ELISAs (all purchased from R & D Systems). The experiments were all performed per the manufacturer’s instructions. All samples were run at a cell homogenate protein concentration of 300 µg/mL. The exposure concentration range selected was 0 and 31.25–500 µg/mL TiO2NPs.

### 2.9. Angiogenesis Proteome Profile

The angiogenesis proteome profiler (R & D Systems) contained 4 membranes, each spotted in duplicate with 55 different angiogenesis antibodies. The assay was performed per the manufacturer’s instructions. This was a qualitative assay that indicates the relative expression of the angiogenesis markers. The concentrations selected were 0 and 100 µg/mL TiO_2_NPs. The concentration selected represented the control and the no observed adverse effect level (NOAEL) (100 µg/mL TiO_2_NPs). The membranes were subjected to an ultra-sensitive chromogenic 3,3′,5,5′-Tetramethylbenzidine (TMB) substrate (Thermo Fisher, Waltham, MA, USA) to show sample–antibody complexes labelled with streptavidin-horseradish peroxidase (HRP). Pictures were taken of the membranes after substrate exposure.

### 2.10. Statistical Analysis

All experiments were performed in triplicate and the data calculated using Microsoft Excel. The data are represented as mean ± standard deviation (SD). A one-way analysis of variance (ANOVA) using SigmaPlot 12.0 (Systat Software Inc., San Jose, CA, USA) was used to determine statistical differences, with *p* < 0.01 deemed significant.

## 3. Results

### 3.1. Nanoparticle Characterization

The TEM images showed the morphology to be spherical and the size to be 21 nm of TiO_2_NPs (Figure 1a). The SEM with energy dispersive X-ray was used to confirm elemental titanium (Ti) and surface morphology (Figure 1b,c). The XRD pattern of TiO_2_NPs (Figure 2) matched the International Centre for Diffraction Data (ICDD) of two mineral forms of TiO_2_ namely, anatase (Power Diffraction File (PDF) card no.: 21-1272) and rutile (Power Diffraction File (PDF) card no.: 21-1276). The (XRD) pattern of TiO_2_NPs (Figure 2) matched the standard diffraction pattern of two mineral forms of TiO2 namely; anatase (JCP card no.: 21-1272) and rutile (JCP card no.: 21-1276). The crystallinity of the synthesized TiO_2_NPs was well-defined, with the narrow and high-intensity XRD peaks indicating large particle sizes. The pattern showed the presence of favourable orientation of planes (101), (110), (004), (200), (105), and (211) which are peculiar to TiO_2_. The (XRD) also showed the different peaks pattern of TiO_2_NPs (Figure 2).

### 3.2. Nanoparticle Behaviour in Various Medias

The NPs in the presence of a 150 mM NaCl solution indicated a steady increase in size across the period assessed. The hydrodynamic size over the 2-week incubation period significantly increased (*p* < 0.001) from 297 ± 43.55 to 647.53 ± 56.18 nm (14 days) (Table 1). However, the zeta potential for the same period exhibited a notable decrease in NP charge at 14 days. It decreased from an initial −23.3 ± 0.66 (0 h) to −12.43 ± 0.45 mV (14 days) (Table 2).

When the TiO_2_NPs were incubated in the presence of PBS, it was noted that size did not change across the 24 h period. It was also notable that the size of the NPs at 0 h were the same for PBS and DMEM in the absence and presence of serum (787.33 ± 65.24 nm). Subsequently, TiO_2_NPs in PBS for 14 days exhibited a significant increase (*p* < 0.014) in size from 787.33 ± 65.24 (0 h) to 1055.8 ± 86.95 nm (14 days) (Table 1). Nonetheless, the zeta potential did not reflect this. The surface charge notably increased (*p* < 0.001) from −26.03 ± 2.29 to −16.05 ± 2.77 mV for PBS over the 24 h period. It then decreased to −27.87 ± 1.97 mV at day 7 and then another significant increase (*p* < 0.001) at day 14 with a charge of −16.48 ± 1.29 mV. Thus, 0 h and 7 days had similar surface charges and 24 h and day 14 have comparable zeta potentials for PBS (Table 2).

The TiO_2_NPs in the presence of DMEM without serum displayed a notable increase (*p* < 0.014) in size from 0 h to 7 days, with sizes ranging from 787.33 ± 65.24 to 1442.35 ± 491.99 nm respectively (Table 1). The surface charge of the NPs was stable after 24 h. The zeta potential increased (*p* < 0.014) from −14.1 ± 0.87 to −8.29 ± 1.9 mV between 0 h and day 7 respectively (Table 2).

TiO_2_NPs exposed to DMEM in the presence of serum showed a similar trend to the NPs in 150 mM NaCl, where there was an increase in particle size over time. The size increased (*p* < 0.003) from 787.33 ± 65.24 to 1121.26 ± 216.6 nm at day 7 and another subsequent increase (*p* < 0.003) to 1821.8 ± 450.9 nm after 14 days (Table 1). Surface charge remained consistent and stable after exposing the NPs to complete DMEM for 7 days even though particle size increased. However, zeta potential of the TiO_2_NPs notably increased (*p* < 0.014) at day 14 (−16.45 ± 1.1 mV) compared to incubation period 0 h to 7 days (Table 2).

### 3.3. Cell Viability

The XTT assay used to evaluate potential cytotoxicity of TiO_2_NPs after 48 h, indicated no effect on viability after the cells were exposed to the NP range assessed in this study (Figure 3).

### 3.4. Inflammatory Biomarkers

The exposure of Caco-2 cells to TiO_2_NPs for 48 h resulted in a significant increase (*p* < 0.001) in NO released from the cells at concentrations ≥62.5 μg/mL TiO_2_NPs (Figure 4a). However, the increase would not be deemed significant in a physiological system. Contrary to the NO data, the other inflammatory markers (i.e., IL-6 and IL-8) secretion levels from the cells remained unaffected after a 48 h exposure period (Figure 4b,c). IL-6 levels secreted by the cells were approximately 3 times higher compared with IL-8 secretion.

### 3.5. Cell Stress Biomarkers

The cell stress biomarkers phospho-HSP-27 and HSP-70 notably increased (*p* < 0.001) at concentrations ≥31.25 μg/mL TiO_2_NPs. Phospho-HSP-27 levels doubled at 31.25 μg/mL TiO_2_NPs and tripled at 125 μg/mL TiO_2_NPs, compared to the 0 μg/mL TiO_2_NP control, and then increased to 7-fold at 250 and 500 μg/mL TiO_2_NPs (Figure 5a). However, with regards to the HSP-70 levels, there was a notable dose-dependent increase in the secretion of this biomarker (Figure 5b). HSP-70 levels were also noted to be a 1000× higher in comparison with the phospho-HSP-27 levels.

Dissimilar to the other cell stress biomarkers, the production of SOD-2 was only considerably upregulated (*p* < 0.001) at 250 μg/mL TiO_2_NPs, with an approximate 1.8-fold increase in its production in comparison to the control (Figure 5c).

### 3.6. Angiogenesis Proteome Profile

The control (0 μg/mL TiO_2_NPs) and NOAEL (100 μg/mL TiO_2_NPs) supernatants assayed for potential angiogenic biomarkers revealed the same proteins except persephin, which was only evident on the control membrane (Figure 6). Based on the intensity of the dot, the proteins dipeptidyl peptidase IV (DPPIV) and endostatin was more prominent on the NOAEL membrane (Figure 6b). However, platelet-derived growth factor AA (PDGF-AA) and angiopoietin-2 was more noticeable in cells exposed to the control concentration (Figure 6a).

## 4. Discussion

In recent years, the harmful effects of TiO_2_NPs on human health have resulted in different perspectives due to the available data. The polymorphic structure and properties of TiO_2_NPs may alter the toxicity of TiO_2_NPs. The potential properties, inertness and wide range of applications have raised concerns regarding the toxicity and safety of TiO_2_NPs [26]. It has been recently classified as a potential carcinogenic factor from group 2B by the international agency for research on cancer (IARC) due to the experimental tests done on animals concerning exposure by inhalation [6,26]. Toxicological in vitro studies revealed a number of harmful effects on mammalian cells not limited to decreased cell viability, DNA damage, increased ROS generation, inflammation and genotoxicity [27].

The physical characteristics of TiO_2_NPs were analysed with TEM, SEM and XRD to determine their sizes and morphology (Figure 1 and Figure 2). The TEM, SEM and XRD analyses showed that the sizes correspond with the manufacturer size of average particle size of 21 nm, and spherical shape. The findings also corroborated the analysis previously done by Romanello and de Cortalezzi 2013 [28], who reported that the NPs forms aggregates of 200–300 nm in aqueous solutions, at pH values where agglomeration is favourable and also revealing the peak patterns to be a hydrophilic fumed TiO_2_ mixture of rutile and anatase forms.

The initial size of TiO_2_NPs in complete DMEM was 787.33 ± 65.24 nm (Table 1) which was similar to what was found by another study where the initial hydrodynamic diameter of TiO_2_NP in complete DMEM was 843 ± 69 nm [29]. However, their zeta potential was −7.4 ± 2.5 mV which is much higher compared with the data generated in this study (−13.03 ± 0.85 mV) (Table 2). It is also proposed by other studies that probe ultra-sonication does not easily break down agglomerates but in fact endorses agglomeration due to the enhanced particle-particle interaction [29,30]. In conjunction with probe sonication, the high ionic strength present in the culture media can also promote agglomeration [29]. This was evident as after 24 h the particle size was 911.67 ± 75.06 nm and 1121.26 ± 216.26 nm after a 7-day incubation period, respectively. This trend was consistent with all the medias in which TiO_2_NPs was evaluated and proposes that the NPs were stable after 48 h, the time frame used in this study. The zeta potential remained mostly stable after size increase in all medias used (Table 2). A study where different reducing agents, namely glycine (gly-TiO_2_NP) and L-alanine (ala-TiO_2_NP), was used to evaluate how complete DMEM impacted the size and surface charge of TiO_2_NP. The authors of the aforementioned study noted an increase in size when placed into cell culture media for 24 h, with hydrodynamic size being 1842.6 ± 263 nm and 1296 ± 662 nm for gly-TiO_2_NP and ala-TiO_2_NP, after an initial size of 85.5 and 72.8 nm respectively [23]. The surface charge after being exposed to complete DMEM with gly-TiO_2_NP and ala-TiO_2_NP was −7.9 ± 0.4 and −8.2 ± 0.2 mV respectively. These results imply that the reducing agent does not impact size once in culture medium as can be seen when comparing the data found in this study.

The TiO_2_NPs were not cytotoxic to the cells at the assessed concentrations (Figure 3). This could be attributed to the increase in NP size (Table 1), as this could impact cellular uptake. This has been investigated by numerous studies, looking at various types of nanoparticles and cells. Studies have ascertained that an increase in particle size would impact cellular uptake and in turn decrease the level of toxicity [17,31,32,33,34]. Several questions remain as a limited number of nanoparticles have been tested, as well as difficulty comparing data reported due to disparities in surface chemistry, purity and size uniformity of the nanoparticles used [34,35,36]. Uptake of these NPs require further investigation to corroborate the proposed mechanism for the lack of cytotoxicity. However, other studies have found similar results with regards to TiO_2_NPs not being cytotoxic to human colon carcinoma cells [35,36,37]. The lack of cytotoxicity could also be attributed to the induction of cell stress biomarkers (Figure 5a–c). As phospho-HSP-27, HSP-70 and SOD-2 are produced to aid in protecting the cells from damage such as hypoxia and cytotoxic exposure [38,39,40,41]. These systems regulate reactive oxygen species (ROS) formation and protect biological systems from ROS induced oxidative damage [42]. In contrast to what we have found, other studies have found that TiO_2_NPs induce cyto-and genotoxicity [11,15,19]. This disparity can be attributed to NP size, cell type and dosage.

The TiO_2_NPs were found not to affect the production of inflammatory cytokines, IL-6 or IL-8 (Figure 4b,c). As constant exposure to nanoparticles that trigger inflammation could result in autoimmunity. The IL-6 data generated agreed with other studies that found that Caco-2 cells exposed to TiO_2_NPs did not affect the expression levels of this cytokine [43,44]. The IL-8 data is different from other studies that investigated the effect of TiO_2_NPs on Caco-2 cells IL-8 expression levels. Kruger et al. (2014) elucidated that TiO_2_NPs activates IL-8 and IL-8-related pathways [43]. This was confounded by other studies, which indicated that exposure of human endothelial cells to TiO_2_NPs resulted in an increase of IL-8 levels after a 24 h exposure period [44,45]. This trend of increasing IL-8 levels was also evident when Caco-2 cells were exposed to nanosilica [34]. An in vivo study looking at inflammatory cytokines such as IL-6 and IL-8 secretion from the small intestine upon TiO_2_NPs exposure found no change in their expression levels, which is in agreeance with our data. However, they did find an increase in other inflammatory cytokines such as IL-4, IL-12, TNF-α and IFNγ [46].

The angiogenesis proteome profile revealed very little differences between the control and 100 μg/mL TiO_2_NPs (Figure 6). There was an inhibition of the protein and pro-angiogenic molecule, persephin upon exposure to 100 μg/mL TiO_2_NPs. However, the protein, DPPIV was more prominent in the NOAEL exposure compared to the anti-angiogenic molecule, endostatin which was also upregulated at the same exposure concentration. The other pro-angiogenic proteins, PDGF-AA and angiopoietin-2 were higher in the control compared to NOAEL. The inhibition of persephin, and suppression of PDGF-AA and angiopoietin-2 in the NOAEL exposure indicates the potential anti-angiogenic effects of TiO_2_NPs, as the activation of persephin, angiopoetin-2 and PDGF-AA triggers the angiogenic process [47,48,49,50]. This is further supported by the upregulation of endostatin in the 100 μg/mL TiO_2_NPs exposure as this is produced endogenously and inhibits angiogenesis [51]. The upregulation of DPPIV in the NOAEL exposure is not significant as this protein is constitutively expressed on endothelial cells [52,53]. These results are supported by Jo et al., who found TiO_2_NPs to be anti-angiogenic in vitro as there was a reduction in vascular endothelial growth factor (VEGF) [50]. Identifying these potential biomarkers is important as angiogenesis promotes the development of new blood vessels from existing ones [51]. This has a direct relation to cancer as tumour growth and metastasis relies on the initiation of angiogenesis and lymphangiogenesis, which can be triggered by chemical signals [54]. However, this requires further in-depth investigation as the size and exposure period of the nanoparticle might impact these factors.

## 5. Conclusions

Caco-2 cells express anti-angiogenic markers (i.e., persephin, angiopoetin-2, PDGF-AA and endostatin) upon exposure to 100 μg/mL TiO_2_NPs and could be a potential candidate for use in cancer therapy. This, with the lack of inflammatory cytokine production and cytotoxicity makes this NP a good candidate. However, further comprehensive investigations are required as the exposure period, dosage and size of the NP could greatly impact the above-mentioned factors.

## Figures and Tables

**Figure 1 biomolecules-12-01334-f001:**
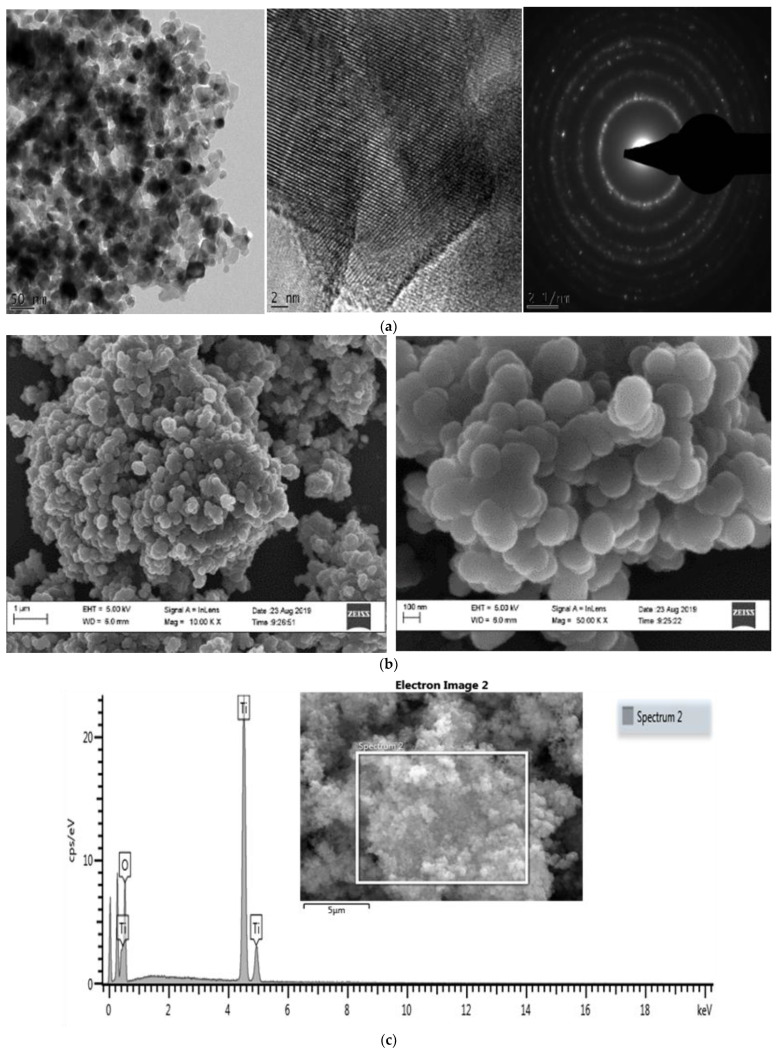
Characterization of TiO_2_NPs; (**a**) TEM analysis of TiO_2_NPs. (**b**) SEM analysis of TiO_2_NPs. (**c**) SEM Energy Dispersive X-ray analysis showing elemental TiO_2_ and electron image analysis of TiO_2_NPs.

**Figure 2 biomolecules-12-01334-f002:**
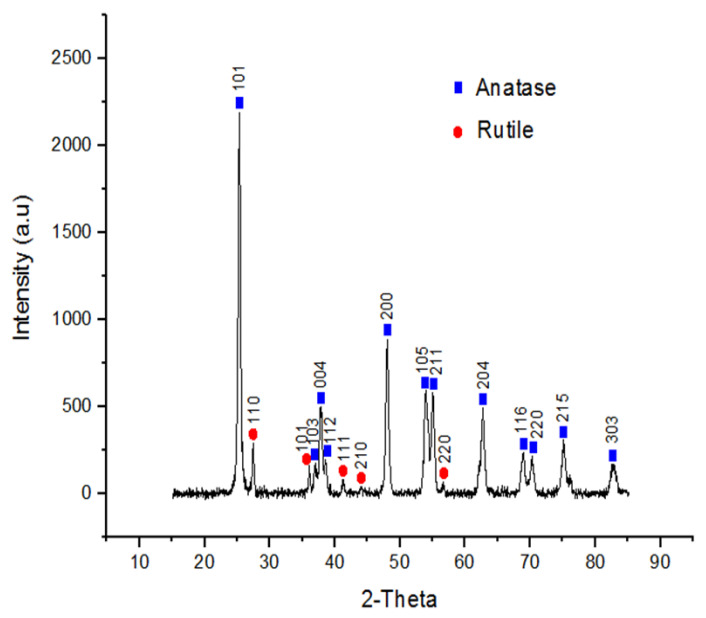
X-ray crystallography (XRD) analysis of TiO_2_NPs.

**Figure 3 biomolecules-12-01334-f003:**
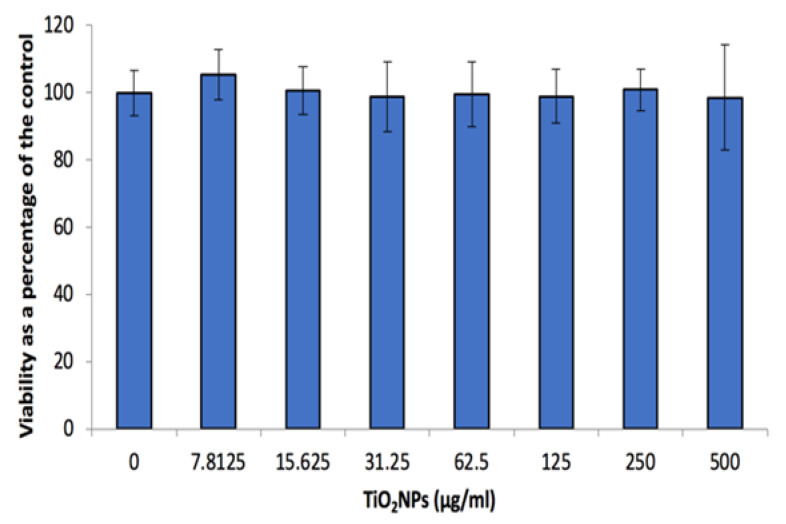
Caco-2 cell viability after 48 h exposure to a range of TiO_2_NP concentrations. Data represented as mean ± SD.

**Figure 4 biomolecules-12-01334-f004:**
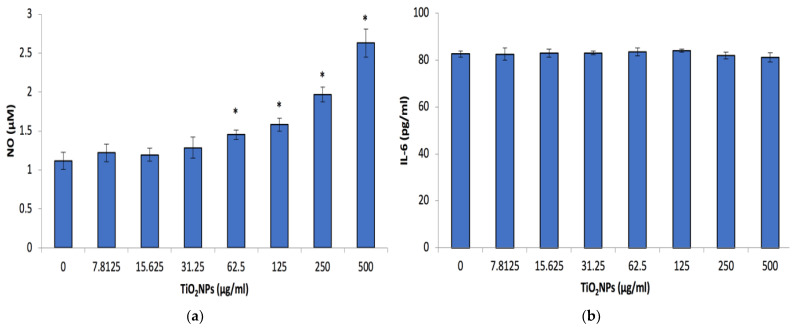
Caco-2 Nitric oxide (NO), Interleukin 6 (IL-6) and Interleukin 8 (IL-8) (**a**–**c**) respectively after 48 h exposure to a range of TiO_2_NP concentrations. Data represented as mean ± SD. Significance demarcated by *, indicating a significant difference of *p* < 0.001.

**Figure 5 biomolecules-12-01334-f005:**
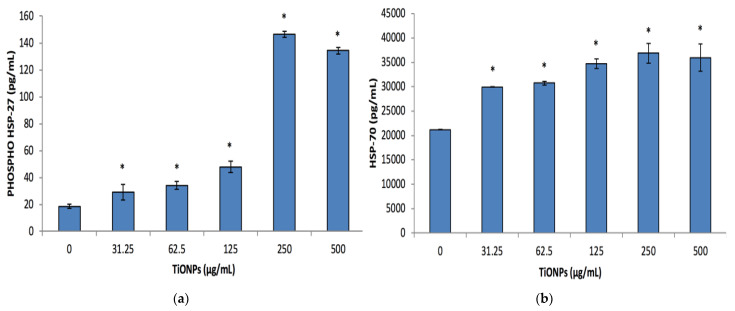
Phospho-HSP-27, HSP-70 and SOD-2 (**a**–**c**) respectively cell stress biomarkers after exposing Caco-2 cells to various TiO_2_NPs, concentrations for 48 h. Data represented as mean ± SD. Significance demarcated by (*), indicating a significant difference of *p* < 0.001.

**Figure 6 biomolecules-12-01334-f006:**
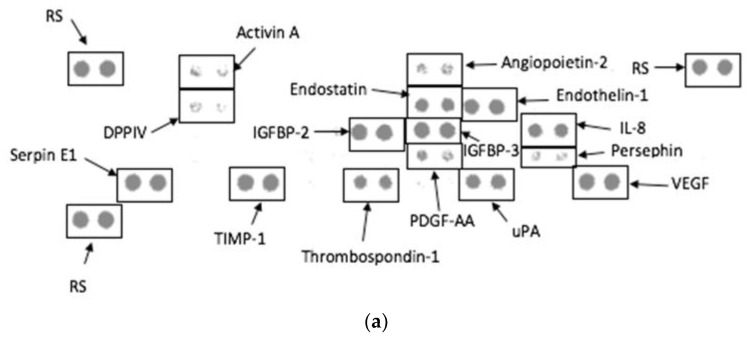
Angiogenesis proteome profile using culture supernatants of Caco-2 cells exposed to (**a**) control (0 µg/mL TiO2NPs) and (**b**) NOAEL concentration (100 µg/mL TiO_2_NPs) for 48 h. RS-Reference spot.

**Table 1 biomolecules-12-01334-t001:** Hydrodynamic diameter (nm) of TiO_2_NPs after being exposed to various physiological medias over a 2-weeks period.

MEDIA	Number of Days			
	0	1	7	14
**150 Mm NaCl**	297.33 ± 43.55	431.87 ± 59.81	347.94 ± 72.48	647.53 ± 56.18 *
**1× PBS**	787.33 ± 65.24	787.33 ± 65.24	915.3 ± 113.18	1055.8 ± 86.95 ****
**DMEM**	787.33 ± 65.24	962 ± 140.63	1442.35 ± 491.99 ****	685.04 ± 121.19
**DMEM (10% FBS)**	787.33 ± 65.24	911.67 ± 75.06	1121.26 ± 216.26	1821.8 ± 450.9 **

Data expressed as mean ± SD. Significance demarcated by (*) indicating significant difference of *p* < 0.001, (**) *p* < 0.003, and (****) *p* < 0.014 compared to the relative 0 h control.

**Table 2 biomolecules-12-01334-t002:** Zeta Potential (mV) of TiO_2_NPs after being exposed to various physiological medias over a 2-weeks period.

MEDIA	Number of Days			
	0	1	7	14
**150 Mm NaCl**	−23.3 ± 0.66	−23.82 ± 1.49	−20.06 ± 5.84	−12.43 ± 0.45 ***
**1× PBS**	−26.03 ± 2.29	−16.05 ± 2.77	−27.87 ± 1.97	−16.48 ± 1.29 *
**DMEM**	−14.1 ± 0.87	−13.76 ± 1.66	−8.29 ± 1.9 ****	−12.62 ± 2.41
**DMEM (10% FBS)**	13.03 ± 0.85	−13.4 ± 1.35	−13.7 ± 1.18	−16.45 ± 1.1 ****

Data expressed as mean ± SD. Significance demarcated by (*) indicating significant difference of *p* < 0.001, (***) *p* < 0.007 and (****) *p* < 0.014 compared to the relative 0 h control.

## Data Availability

Not applicable.

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
