# Peer review of "The Stability and Anti-Angiogenic Properties of Titanium Dioxide Nanoparticles (TiO2NPs) Using Caco-2 Cells"

_biomolecules, 2022, doi:10.3390/biom12101334_

Round 1

Reviewer 1 Report

Review on the manuscript entitled “Stability and Anti-Angiogenic Properties of Titanium Dioxide Nanoparticles (TiO2NPs) Using Caco-2 Cells” by Oladipupo Moyinoluwa David et al.

For some years now, the toxicity and the pollution done by TiO2 accumulation was intensively studied. Even with some negative inputs, TiO2 is still used indeed with caution in some applications, but there still are some aspects that must be clarified by further research.

Therefore, the research subject is interesting, but the manuscript must undergo considerable improvement.

First, I want to mention that the manuscript was hard to understand because of the English and the lack of flow in the ideas. The abstract and introduction must be written again.

Line 66 – “This was confirmed by further testing [26]” -  Please remove the sentence. The authors further provided the equipment that was used in order to characterize the NPs.

For the NPs provided by Evonik Degussa Corporation, please specify lot/batch/number/data sheet.

Line 72 – “These NPs were subsequently further characterized an…. is favourable [26]” - should be moved in the discussion part and not mixed with the characterization methods.

Line 93 - 2.2. Nanoparticle Preparation for Cell Culture – repeats the same text as lines 76-78

Section 2.3. – Please mention the standard tissue culture conditions

3.1. – The pictures from TEM and SEM must have a readable scale. All the pictures and figures must be renumbered, at this moment creates confusion and is hard to follow. Please merge the TEM, SEM and EDX in one figure – Figure 1 and leave the X-Ray diffraction in a different figure. In addition, the XRD pattern must include a DB card or PDF identification card that could indicate the attribution of peaks. The Miller indices (for TiO2?) are present on most of peaks but there are some that are not allocated, please explain.

3.2. Nanoparticle Characterization in Various Medias – Should be named Nanoparticles Behavior in Various Medias. Furthermore, all the results presented in text are present in Table 1, so please use only one form for presenting your results. In Table 1 – the measured parameters should be added - "Hydrodynamic diameter" in nm that is specified only in the caption.

Regarding the results presented in Table 1 there are some unexpected inconsistencies, for example 150 Mm NaCl in day 7 the hydrodynamic diameter decreases. I searched for an explanation in the discussion part but I did not find one, so please explain these type of changes in terms of hydrodynamic diameter and zeta potential.

Line 273 - Please renumber the figure following the previous section and avoid duplicate text. The viability at 7.8125µg/ml is above 100%? Please explain.

Figure 3 a, b, c should be merged, not presented separately

Figure 4 a, b, c should be merged, not presented separately

Figure 4 a and 4 c- Why the 250 µg/ml presents a higher stress biomarker than 500 µg/ml?

Is hard to understand Figure 5 due to the poor contrast, please put a figure that is clearer.

Line 429 - The zeta potential remained mostly stable after size increase in all medias used (Table 2) – Please verify this affirmation considering the stability domain.

I kindly ask the authors to revise their manuscript according to the above comments.

Reviewer 2 Report

This manuscript focused on the effects of various medias on TiO2NP hydrodynamic size and zeta potential. In addition, the effects of these characterized TiO2NPs on Caco-2 cell viability, cell stress and inflammatory biomarkers as well as angiogenesis proteome profile were investigated. Some comments are given here.

1. The characterization of physical and chemical properties of TiO2NPs was well done. But the results were less described. Such as particle size distribution, crystal form ratio and specific surface area were not described in detail. The figures of relevant results lack necessary descriptions or annotations to facilitate readers to obtain key information.

2. The effects of various medias on TiO2NP hydrodynamic size and zeta potential were studied in detail. What is the relationship between these various medias and subsequent cell assay studies? Please explain the biological significance of these data.

3. The results of cytotoxicity of TiO2NPs were different from most similar studies. No obvious cytotoxicity was found after exposure to 500 μg/ml TiO2NPs for 48 h, which seems to be a little inconsistent with expectations. Did the author conduct a repeat experiment? If it is determined that such results are accurate, please explain the possible reasons.

4. Why did these cell stress biomarkers such as Phospho-HSP-27, HSP-70 and SOD-2 decrease at dose of 500 μg/ml? In addition, these figures should be merged.

5. It is suggested that some quantitative analysis results should be supplemented on the assay for potential angiogenic biomarkers.

6. The effects of TiO2NPs on Caco-2 cell viability, cell stress and inflammatory biomarkers as well as angiogenesis proteome profile were investigated. The relationship between these effects should be properly summarized or discussed.

7. Table 1, Table 2 and Figure 3c, The marking of significant difference was not professional enough. It is suggested to unify the type and interpretation of annotations.

8. There are some old references in the reference section, and it is suggested to update them to those in the past 5 years. In addition, some references were repeated, such as 2 and 40. It is recommended to check carefully.

Round 2

Reviewer 1 Report

Second review on the manuscript entitled “Stability and Anti-Angiogenic Properties of Titanium Dioxide Nanoparticles (TiO2NPs) Using Caco-2 Cells” by Oladipupo Moyinoluwa David et al.

I previously asked the authors to improve and rewrite the abstract and the introduction, unfortunately in the introduction I do not see the requested modification, still lacks the flow in ideas and the information is not presented well. Please improve the English… In addition, the introduction should briefly present the angiogenesis regarding the presented subject.

On Figure 1a the scale should be put in more visible way, at this point is hard to read. I already asked this in my first evaluation; please see “The pictures from TEM and SEM must have a readable scale”

Figure 2a and 2b could be unified, on every diffraction peak can be put the allocation (red/blue color) and afterwards the Miller indices. Also, please put the complete name for the used cards “JCPDS card no.”.

Line 213, 231 – Please do not use repeatedly: X-ray diffraction (XRD), transmission electron microscope (TEM) , Scanning electron microscope (SEM) – All the characterization techniques are described in section 2.1.

Regarding the information in Table 1, as I previously asked, please explain why in day 7 the hydrodynamic diameter is not consistent with the values from day 1 and 14. In my opinion, the values should keep an increasing tendency.  Please explain the variation for Table 2 as well.

How this variation in particle size is correlated with the viability for example?

Line 327 – please remove Figure 2 after Figure 3 and check carefully all the manuscript.

At my previous question regarding the viability, the authors replied, “The viability at 7.8125µg/ml is above 100% because the NPS was not cytotoxic to the cells as seen across the concentration. But the slight increase at 7.8125µg/ml was insignificant.” So, is to be understood that not only the TiO2 is not toxic but also slightly stimulates proliferation?

One of the answer of the authors was “At 250 µg/ml the activity of the cell stress biomarkers are higher and does not affect toxicity but the decrease observed at 500 µg/ml was not significant, which could possibly be as a result of biological variations.” – For me is not clear yet, if can be said that the difference is possible due to the biological variation…then all the experimental data should be affected? How many times was every experiment done?

Another issue will be the connection between the determination of parameters for the TiO2 NPs and the cell culture. What is the connection between the two parts?

Reviewer 2 Report

No other comments.

Author Response

We appreciate the reviewer for his/her helpful comments and suggestions.